# Classification and Identification of Industrial Gases Based on Electronic Nose Technology

**DOI:** 10.3390/s19225033

**Published:** 2019-11-18

**Authors:** Hui Li, Dehan Luo, Yunlong Sun, Hamid GholamHosseini

**Affiliations:** 1School of Information and Engineering, Guangdong University of Technology, Guangzhou 510006, China; 1111503005@mail2.gdut.edu.cn (H.L.); dehanluo@gdut.edu.cn (D.L.); 2School of Electric and Automatic Engineering, Changshu Institute of Technology, Changshu 215500, China; 3School of Engineering, Computer and Mathematical Sciences, Auckland University of Technology, Private Bag 92006, Auckland 1142, New Zealand; hamid.gholamhosseini@aut.ac.nz

**Keywords:** electronic nose, industrial gas, classification and identification, kernel discriminant analysis

## Abstract

Rapid detection and identification of industrial gases is a challenging problem. They have a complex composition and different specifications. This paper presents a method based on the kernel discriminant analysis (KDA) algorithm to identify industrial gases. The smell prints of four typical industrial gases were collected by an electronic nose. The extracted features of the collected gases were employed for gas identification using different classification algorithms, including principal component analysis (PCA), linear discriminant analysis (LDA), PCA + LDA, and KDA. In order to obtain better classification results, we reduced the dimensions of the original high-dimensional data, and chose a good classifier. The KDA algorithm provided a high classification accuracy of 100% by selecting the offset of the kernel function *c* = 10 and the degree of freedom *d* = 5. It was found that this accuracy was 4.17% higher than the one obtained using PCA. In the case of standard deviation, the KDA algorithm has the highest recognition rate and the least time consumption.

## 1. Introduction

With the rapid development of industrial technology, various flammable, explosive, and toxic dangerous gases are widely produced. In industrial production, continuous monitoring of such gases is an important and challenging task for enterprise safety, especially in the power, coal, and petrochemical industries [1]. Toxic gases play an important role in environmental pollution [2]. Therefore, timely and accurate detection and monitoring of these harmful gases are of great value [3]. Industrial gases are widely spread around metallurgy, steel, petroleum, chemical, machinery, electronics, glass, ceramics, building materials, construction, food processing, and other industries [4,5,6,7,8,9,10].

Fortunately, advanced gas detection technologies, such as the portable electronic nose (E-nose), have become widely available to tackle the issues. Some researchers have applied E-nose technology to inspect the quality of food, including fruits [11,12], meat [13,14], rice [15,16], tea [17], cigarettes [18], Chinese herbal medicines [19], and others plant crops [20]. The latest reports show that an E-nose can be applied to the process of harmful gas detection [21,22,23], the medical industry [24,25], and space applications [26,27,28]. Because of this, more investigations into E-nose sensors and gas-chromatography/mass spectrometry (GC/MS) applications have been published [29,30].

The E-nose system consists of gas sensors and signal processing and pattern recognition components [31]. It attempts to simulate the perception, analysis, and judgment of odor in human/animal olfactory organs [32,33,34,35,36,37,38,39]. Compared with conventional odor analysis techniques, E-nose technology has the advantage of reflecting the “odor characteristics” of the gas being measured. It is fast, sensitive, accurate, and nondestructive and has more objective and repeatable characteristics than traditional artificial odor recognition. However, as a result of the variety of its sensors, and its erratic accuracy and reliability, there are challenges in employing E-nose technology for industrial gas identification. Therefore, it is necessary to apply preprocessing methods before extracting the features of the gas data. More specifically, applying proper signal processing and pattern recognition methods can result in better and more accurate detection and identification of the industrial gases.

In this study, an E-nose was used to detect the odor information of four different typical industrial gases, carbon dioxide (CO_2_), methane (CH_4_), ammonia (NH_3_) and volatile organic compounds (VOCs) [40,41,42]. We performed detailed calculations and used a precise experimental setup for the data collection using the E-nose. Typical methods such as principal component analysis (PCA), linear discriminant analysis (LDA), a support vector machine with radial basis function (RBF) kernel (SVM-rbf) [43], a support vector machine with sigmoid kernel (SVM-sig) [44], and a back-propagation (BP) neural network [45] were used to reduce the dimensionality and feature extraction of high-dimensional data collected by the E-nose. Further, a linear classification algorithm based on kernel function is proposed. It has a better performance than traditional linear discriminant methods through theoretical calculations and has been verified by experimental results. The proposed method obtained better classification and recognition results for different kinds of industrial gases by optimizing the classifier parameters.

## 2. Experimental Setup and Methods

### 2.1. Experimental Setup

#### 2.1.1. Electronic Nose

In this paper, the selected device for collecting the industrial gases is a portable electronic nose (PEN3 E-nose). It was manufactured by AIRSENSE, Schwerin, Germany (https://airsense.com/en/products). In order to improve the stability of the object detection, 10 metal oxide gas sensors are used in the PEN3 to form the acquisition array and improve the reliability and range of detection. The recorded data are called the odor data map (ODM). The name of the sensor array and related parameter information are shown in Table 1.

#### 2.1.2. Experimental Setup

An industrial gas detection experimental setup is shown in Figure 1. Firstly, the gas samples were stored in a reinforced steel bottle and the released gas flow was adjusted by the pressure valve. We obtained the specific concentration gas samples using a gas distribution device (GDD). The gases were fluxed alone from the gas channels. We used air as carrier gas during the experiment. Air was used to clean the GDD and PEN3 during the experiment. The effect of carrier gas on E-nose response was negligible and so was ignored. Then, when the gas inflow reached the electronic nose sampling interface, the sensor array responded and recorded the gas signal. The saved ODM can be used for further feature extraction and analysis using a pattern recognition method. Finally, the identified results were presented as the output of the detection system. The Appendix A (Appendix A) is a supplementary description of the experimental setup.

In this paper, the four typical industrial gases selected in the same test environment were carbon dioxide (CO_2_), methane (CH_4_), ammonia (NH_3_), and volatile organic compounds (VOCs). The dynamic headspace sampling method was used to collect samples of the gases by PEN3 E-nose. Each type of gas sample was continuously collected and measured in 31 measuring samples, with a total of 124 sample groups being stored (Table 2). We used 25 sample groups of each gas type for training (with a total of 4 × 25 = 100 training sample groups), the remaining six sample groups were used for testing (with a total of 4 × 6 = 24 testing sample groups). The sampling period was 120 s and ODM (the response value of the sensor) was recorded every second. Therefore, the number of samples collected for each group was 120 and the total collecting samples for each sensor was (4 × 25 × 120 = 12,000). Considering an array of 10 sensors in the PEN3, the original size of the training matrix was (12,000 rows, 10 columns). The specific gas samples and experimental setup information are shown in Table 2.

We measured the gas samples at room temperature and other room conditions (the temperature of 25 ± 1 °C, relative humidity of 60% ± 10%). The sampling period was 120 s and the intake airflow when sampling was set to 7.8 mL/s. The response value of the sensor was recorded every second. The sample data of a single sample is a data matrix of 120 rows and 10 columns. In this paper, the steady-state response value is considered to be between the 45th and 57th seconds of the sensor response. The acquired response map is a relatively steady-state process that reflects the validity of the sampled data. The mean value is the average response data value of the sensor’s 120 s sampling time.

#### 2.1.3. E-nose Response of the Selected Four Industrial Gases

We obtained a typical four-gas response map and a response radar map of the gases passing through an E-nose at a concentration of 300 ppm, as shown in Figure 2.

The figure clearly shows the responses of 10 sensors to different gas intensities of CO_2_, CH_4_, NH_3_, and VOCs for different periods. The VOCs samples in this paper were derived from indoor VOCs.

As shown in Figure 2, the response characteristics of the radar map are different for some sensors (e.g., S2, S4, S6, S8, and S9). In addition, sensors S2 and S9 are more sensitive to NH_3_ and VOCs than CO_2_ and CH_4_, and sensors S6 and S8 are more sensitive to CO_2_ and VOCs than CH_4_ and NH_3_. There is a high correlation between the detection of the four industrial gases due to the difference in sensor detection accuracy and the detection of cross-sensitivity. To further highlight the response characteristics of each sensor to the detected gas, we converted the response curve of the gas to indicate the detection of the odor fingerprint.

By highlighting the response characteristics of a single sensor in terms of gas detection, it can effectively show the contribution of the PEN3 E-nose to the detection response. The similarities and differences are shown in Figure 2. The response maps of CO_2_ and VOCs are very similar since the sensor is not only sensitive to VOCs, but also responds to CO_2_. This is the characteristic of metal oxide sensors. One of the major objectives of this research was to realize effective features of the gases and the impact of each sensor’s characteristic on the classification of gas samples. After processing of the data acquired by the E-nose, the dimensionality reduction was performed. The influence of data dimension reduction obtained by the sensor array was proposed. At the same time, a semi-supervised embedded kernel classification algorithm was proposed. Next, it is compared with the linear classification algorithm in terms of the recognition rate and algorithm complexity.

### 2.2. Signal Processing Method

#### 2.2.1. An Overview of Principal Component Analysis (PCA)

PCA is a classical linear dimension reduction method based on statistical analysis. It is widely used in data mining, machine vision, and other fields. It is an unsupervised linear mapping transformation used to extract the main feature components of data. Through linear transformation, the original data is represented by statistical feature quantity. The original high-dimensional data is mapped to reduce its dimensionality. Because of the mapping from n-dimensional features to k-dimensional features, the k-dimensional features are a new orthogonal feature. It is a k-dimensional feature reconstructed based on the original n-dimensional features. A set of mutually orthogonal coordinate axes are sequentially sought from the original space. The choice is closely related to the data itself. A set of eigenvectors are sought, which can express the original data as linear polynomials; thereby, realizing the projection occlusion from high dimensionality to low dimensional space.

#### 2.2.2. An Overview of Linear Discriminant Analysis (LDA)

LDA is a supervised learning algorithm. The data is projected in a low dimension using the dimensionality reduction technique. It is hoped that the projection points of each type of data are as close as possible, and the distances between the type centers of different types of data are as big as possible. We can determine the category based on the position of the new sample projection point when classifying a new sample. This specific algorithm can be described as follows:

Sample mean vector $m_{i}$:
(1)$$m_{i}=\frac{1}{N_{i}}\sum_{x\in G_{i}}x,i=1,2,\cdots N.$$
where $m_{i}$ is the mean feature vector of training samples in *i*th class? The number of known pattern classes is N, as $G_{1},G_{2},\cdots G_{N}$, the pattern $x\in R^{n}$ is n-dimensional real vector, and $N_{i}$ is the number of training samples in the *i*th class.

Total within-class scatter matrix $S_{\omega }$:
(2)$$S_{\omega }=\sum_{i=1}^{N}\sum_{x\in G_{i}}(x-m_{i}){\left(x-m_{i}\right)}^{T},i=1,2,\cdots N.$$


In Equation (2), $S_{\omega }$ is the sum of the covariance matrices of similar samples, the variance of the same sample after projection is as small as possible, the sample is as close as possible.

Between-class scatter matrix $S_{b}$:
(3)$$S_{b}=\sum_{i=1}^{N}(m_{i}-m){\left(m_{i}-m\right)}^{T},i=1,2,\cdots N.$$


In Equation (3), $S_{b}$ is covariance matrix of the mean of each class of sample, the sample projection between classes is as far apart as possible to facilitate the reduction of data dimensionality. Therefore,
(4)$$m=\frac{1}{N}\sum_{i=1}^{N}m_{i},i=1,2,\cdots N.$$


The optimal projection direction can be found using the Fisher criterion function defined as:
(5)$$J_{F}(\omega )=\frac{\omega^{T}S_{b}\omega }{\omega^{T}S_{\omega }\omega }.$$


In Equation (5), the ratio of between-class scatter and within-class scatter can be regarded as a comprehensive measure. It determines the data separability after projection. When the full separability measure obtains the maximum, the Fisher optimal projection direction is determined.

#### 2.2.3. Kernel Discriminant Analysis (KDA) Algorithm

In the process of linear discriminant analysis, if the within-class scatter matrix $S_{\omega }$ is a singular matrix, it is impossible to find the best projection. In order to overcome this problem, the method of embedding kernel function is proposed to find an optimal solution by employing kernel discriminant analysis (KDA).

It can be stated that the discriminant rule function is modified by constructing the embedded kernel function. In the Fisher criterion function, the optimal projection cannot be solved because of the singularity of the scatter matrix. According to the LDA, the criterion of the optimal projection is as follows: the within-class scatter matrix is as small as possible, the between-class scatter matrix is as far apart as possible. The discreteness criterion of the construction is as follows:
(6)$$\left\{ \begin{array}{l} D_{F}(\omega )=\frac{\omega^{T}S_{b}-\omega K(x)*\omega^{T}S_{\omega }\omega }{\omega^{T}\omega }=\frac{\omega^{T}(S_{b}-K(x)*S_{\omega })\omega }{\omega^{T}\omega }\\ K(x)={\left[(x_{i}*{x^{\prime }}_{i+1})+c\right]}^{d} \end{array}\right..$$


In Equation (6), the embedded kernel function K(x) is a polynomial function, where c is the offset of the kernel function and d is the degree of freedom. The main function of the K(x) is to balance the $S_{b}$ and $S_{\omega }$, and then the equation can be optimized. The Lagrangian multiplication is applied to Equation (6) to solve the eigenvalue vector of $D_{F}(\omega )$ and get the best projection. Suppose λ is the eigenvalue, then
(7)$$L(\omega ,\lambda )=D_{F}(\omega )-\lambda (\omega^{T}\omega -1)=\omega^{T}(S_{b}-K(x)*S_{\omega })\omega -\lambda (\omega^{T}\omega -1).$$


To find the partial derivatives of the Equation (7), then
(8)$$\frac{\partial L(\omega ,\lambda )}{\partial \omega }=(S_{b}-K(x)\times S_{\omega })\omega -\lambda \times \omega =0,$$
(9)$$(S_{b}-K(x)\times S_{\omega })\omega =\lambda \times \omega .$$


In Equations (8) and (9), λ is the eigenvalues of the matrices formed by $S_{b}-K(x)\times S_{\omega }$.

In summary, the kernel function method avoids the problem whereby the optimal projection cannot be solved effectively due to the singularity of the scatter matrix in the linear discriminant analysis. The implementation steps of the KDA algorithm are described in Algorithm 1:


**Algorithm 1: KDA algorithm**
**Input**:Data matrix $X_{m\times n}$ composed of m samples.
**Step 1:**
Calculate the sample mean matrix ${\bar{u}}_{i}$ of each class, and the overall sample mean matrix u.
**Step 2:**
Calculate the within-class scatter matrix $S_{\omega }$ and the between-class scatter matrix $S_{b}$ for the overall sample.
**Step 3:**
Calculate the optimal projection surface. In Equation (6), the embedded kernel function K(x) is a polynomial function, where c and d are constants, c is the offset of the kernel function, d is the degree of freedom. Calculate the eigenvalue vector $D_{F}(\omega )$ to get the optimal projection. $\lambda ={\left(S_{b}-K(x)\times S_{\omega }\right)}^{-1}$, when we set different values of constant c and d in kernel function K(x), the projection results will be different.**Output**:Reduce result the projection mapping matrix $Y^{\prime }=\lambda \times X_{m\times n}$.

In Setp 1, the samples data is a matrix of m rows and n columns, then
$$X_{m\times n}=\left[\begin{array}{cccc} x_{11} & x_{12} & \cdots & x_{1n}\\ x_{21} & x_{22} & \cdots & x_{2n}\\ \vdots & \vdots & \ddots & \vdots \\ x_{m1} & x_{m2} & \cdots & x_{\mathrm{mn}} \end{array}\right].$$


The sample mean matrix ${\bar{u}}_{i}=\sum_{i=1}^{m}X_{m\times n}$, the overall sample mean matrix $u=\sum_{i=1}^{n}{\bar{u}}_{i}$.

In Setp 2, the within-class scatter matrix $S_{\omega }=\sum_{i=1}^{n}\sum \left(x-u_{i}\right){\left(x-u_{i}\right)}^{T}$, the between-class scatter matrix $S_{b}=\sum_{i=1}^{n}(u_{i}-u){\left(u_{i}-u\right)}^{T}$.

In Setp 3, according to the criterion of the optimal projection with the linear discriminant analysis, the within-class scatter matrix is as small as possible, the between-class scatter matrix is as far apart as possible. For more information on the relationship, refer to Equation (6).

Finally, we select the matrix $Y^{\prime }$ as the dimensionality reduction output, the projection mapping matrix $Y^{\prime }=\lambda \times X_{m\times n}$.

## 3. Results and Discussion

### 3.1. Discrimination between the Selected Four Industrial Gases Using the Four Methods

#### 3.1.1. Classification Results of PCA

PCA can reduce the dimensions and observe the preliminary assessments of similarity between classes. PCA is a projection method that makes it easy to display all the information contained in a dataset.

Figure 3 shows the PCA of the industrial gases in the case of steady-state response and mean value. In the steady-state case, the variance contribution rate of PC1 is 74.66%, and PC2 is 13.75%, then the total contribution rate of the first two PCs is 88.41%. In the mean value case, the variance contribution rate of the PC1 is 74.88%, the PC2 is 13.24%, so the total of the first two PCs contribution rate is 88.12%. In the two-dimensional graph, it can be seen that the distribution of sample points of CH_4_ and NH_3_ is not concentrated in Figure 3a. The intraclass distance is large, except that the points of CO_2_ samples are relatively concentrated. The between-class distance of the four types of samples is small; there is even a case of sample overlap. Therefore, the classification effect of PCA in the steady-state response is relatively poor. The four types of sample points have vast intraclass distances in Figure 3b. The between-class distance is not ideal, the sample coverage is broad, except that NH_3_ samples can be classified with the other samples. Overall, the classification results of PCA in industrial gas samples are not satisfactory.

#### 3.1.2. Classification Results of LDA

The classification results of Figure 4 and Figure 5 are applied to the classification and identification of industrial gas samples by the LDA, and PCA + LDA algorithms, respectively.

As shown in Figure 4, the classification results of the industrial gases are different in the case of steady-state response and mean value. The variance contribution rate of PC1 is 55.78%, and PC2 is 42.36%, the total contribution rate of the first two PCs is 98.14% in Figure 4a. The variance contribution rate of PC1 is 58.53%, and PC2 is 39.82%, so the total contribution rate of the first two PCs is 98.35% in Figure 4b. From Figure 4a, it can be seen that the point of each sample distribution is not concentrated enough, the within-class distance is not small. However, the between-class distance is good, except when the samples of CO_2_ and VOCs have a little overlap. Therefore, the result of classification is poor. As shown in Figure 4b, the classification effect of mean value is not better than in the case of the steady-state response. In general, the classification accuracy of LDA is not high enough and needs to be improved.

Figure 5 has some similarities with Figure 3 and Figure 4. The within-class distance is not small enough, the between-class distance is not large enough. Some parts of the samples have overlapping, the classification results are not good enough. However, the performance of the PCA + LDA algorithms is better than the others in terms of both the total contribution rate and a single contribution rate. The total contribution rate is as high as 99.48% and 99.26%, and even the contribution rate of the PC1 reaches to 91.77% in Figure 5b. Although the total contribution rate of LDA is higher than that of the PCA, the single-axis contribution rate of LDA is second to that of the PCA from Figure 3 and Figure 4. On the basis of the above analysis results, this paper proposes an optimized method to improve the classification and identification of the industrial gases.

#### 3.1.3. Classification Results of KDA

The classification results of the KDA algorithm for four industrial gases are shown in Figure 6 and Figure 7. In this method, the constants c and d were effectively set to fulfill the optimization result. Figure 6 shows the results of the classification of four industrial gas samples in the standard deviation case with different values of ‘c’ and ‘d’. Many experiments were conducted with different constants as shown in Figure 6a c = 10, d = 10; Figure 6b c = 10, d = 7; Figure 6c c = 10, d = 5; Figure 6d c = 10, d = 3; Figure 6e c = 7, d = 5; Figure 6f c = 15, d = 5. From the comparison of the classification results of Figure 6a–c and Figure 5d, the best value of the constant *d* is 5. It can be seen from the comparison of the classification results of Figure 6c,e,f that the best value of the constant c is when c ≥ 10. However, the complexity $O(N^{2})$ (to calculate the matrix of N × N) increases as the value of c increases. In this paper, we selected c = 10 and d = 5 as effective settings after careful consideration. As shown in Figure 6, the classification results are much better than those in Figure 3, Figure 4 and Figure 5. The classification results of each sample from Figure 6 show that the within-class distance was small, the between-class distance was large, and the sample points did not overlap. The classification results of Figure 6c,f were better than the other four classification results. Considering the complexity of the algorithm, Figure 6c shows the optimal classification result of the KDA algorithm under the extracted feature of standard deviation.

On the basis of the classification results of Figure 6 (setting constant c = 10, d = 5), Figure 7 shows the classification results after extracting different features. For example, the Figure 7a differential value; Figure 7b integral value; Figure 7c mean value; and Figure 7d variance value. As shown in Figure 7, the classification results are invalid except for the differential values; the results in the other three characteristics are poor. It can be seen from Figure 7b–d, the within-class distance was large, and the between-class distance was small. The overlap of the sample points was serious, and the classification results were not good. Comparing the classification results of Figure 6c and Figure 7a, we get the best classification results of the four industrial gases via the KDA algorithm with the extracted feature of standard deviation (c = 10, d = 5).

### 3.2. Recognition Accuracy Comparison of Different Classification Methods

The Mahalanobis distance was used to analyze the test sample set, in order to further analyze the classification results of the four industrial gas data sets using various algorithms. The identification results of the test gas samples are shown in Table 3.

The average recognition rate is defined as the average value from all the test samples that correctly recognized the gases. As shown in Table 3, for the 24 test samples, the average recognition rate of PCA was 95.83% because one sample of VOCs was wrongly identified. The average recognition rate of LDA was only 83.33%, wherein the samples of NH_3_ were correctly identified. The average recognition rate of PCA + LDA was good; two types of test samples were wrongly identified. The recognition rates of SVM-rbf and SVM-sig were very poor, only one type of test sample was correctly identified, the remaining sample recognition rates were very low. The recognition rate of BP was relatively well balanced, but it was not ideal. The average recognition rate of KDA was 100%. The test sample set was correctly identified. As we can see from the comparison of the results in Table 3, the recognition rate of KDA was the best; PCA, PCA + LDA, and LDA came second, third, ad fourth, respectively; and the other three methods were far behind.

### 3.3. Time Consumption of Different Classification Methods

We considered the time consumption of the data test by the PCA, LDA, PCA + LDA, and KDA algorithms in order to compare the performance of these algorithms. In this paper, we used MatlabR2014b to analyze and process the high-dimensional data of the four industrial gases. The test hardware environment was CPUi5-6300U, memory 8 GB, main frequency 2.4 GHz. The operation times of the selected algorithms are shown in Table 4.

It can be seen from Table 4 that the time consumption of the selected algorithms for the processing of the four industrial gas samples is generally low. The comprehensive comparison results of the time consumed are as follows: PCA > PCA + LDA > KDA > LDA. The LDA algorithm directly solves the eigenvalues and eigenvectors of the sample matrix. As a result of the nonsingular matrix the time consumption of the LDA algorithm was the lowest. The classification effect of the KDA algorithm is similar in c = 15, d = 5 and c = 10, d = 5, but the latter was better than the former. As a result of the limitation of the data samples in this experiment, the difference was not particularly large in Table 4. However, this difference will be clearly reflected as the sample numbers increase.

## 4. Conclusions

In this paper, the PEN3 E-nose was used to obtain the response curves of different industrial gas samples. The nonlinear high-dimensional odor data were identified and analyzed using various analytical methods.

The classification results of the KDA algorithm were better than the other classification results. The recognition rate of KDA was compared with those of PCA, LDA, PCA + LDA, SVM-rbf, SVM-sig, and BF. However, the KDA classification effect differed under different feature extraction and constant values of the c and d settings (c is the offset of the kernel function and d is the degree of freedom). The final experimental results show that KDA has the best classification performance with c = 10, d = 5 and the extracted feature of standard deviation (Figure 6 and Figure 7). The intraclass distance was small, the between-class distance was large, and the recognition rates of the test samples were the highest at 100%. It was 4.17% higher than those of PCA. The time consumption of the KDA algorithm was the lowest among the selected classifiers except for LDA, and the time consumption differed for different constant values. The results of constant c = 10, d = 5 were better than for c= 15, d = 5 (Table 4). The classification results of the KDA algorithm indicate that it is a feasible and efficient method. The results of this method can be beneficial in industrial gas detection and safety applications. However, gas monitoring at different concentrations and different gas mixtures still requires further studies.

## Figures and Tables

**Figure 1 sensors-19-05033-f001:**
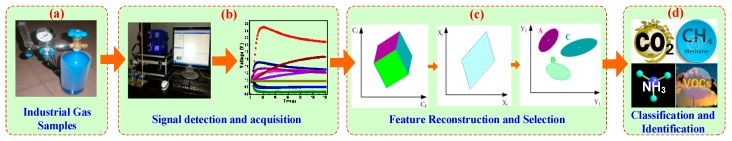
Industrial gas detection and identification experimental setup. (**a**) Gas samples storage and flow control. (**b**) The gas signal is detected and acquired using an E-nose. (**c**) Feature extraction and reconstruction of the gas signal using the signal processing methods. (**d**) The classification and recognition results are presented as an output.

**Figure 2 sensors-19-05033-f002:**
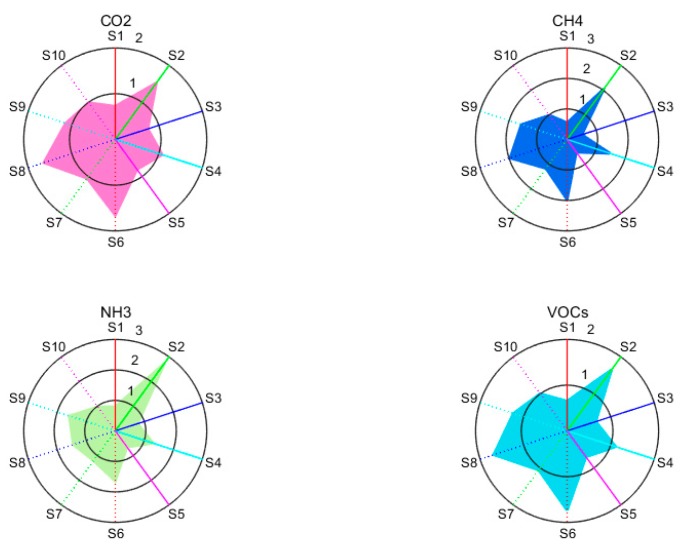
Radar maps of four industrial gas samples: CO_2_, CH_4_, NH_3_, VOCs, from top-left to the bottom-right, respectively.

**Figure 3 sensors-19-05033-f003:**
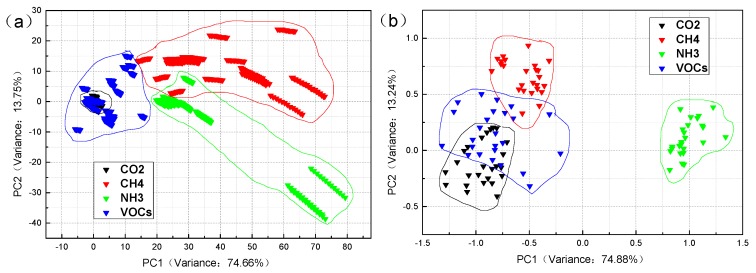
The principal component analysis (PCA) classification results of four industrial gas samples: (**a**) steady-state response value and (**b**) mean value. The Appendix A (Appendix A) shows an enlarged version of the Figure 3a.

**Figure 4 sensors-19-05033-f004:**
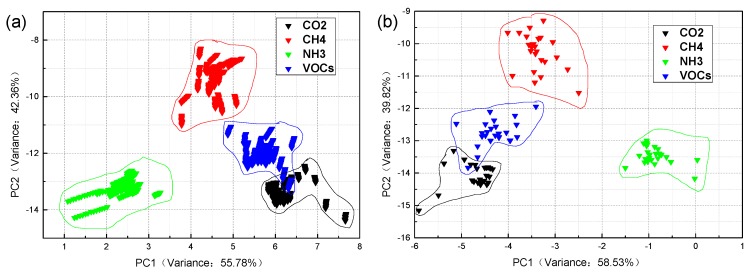
The linear discriminant analysis (LDA) classification results of four industrial gas samples: (**a**) steady-state response value and (**b**) mean value. The Appendix A (Appendix A) shows an enlarged version of the Figure 4a.

**Figure 5 sensors-19-05033-f005:**
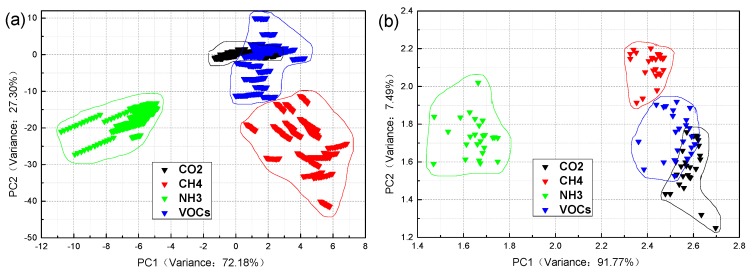
The PCA + LDA classification results of four industrial gas samples: (**a**) steady-state response value and (**b**) mean value. The Appendix A (Appendix A) shows an enlarged version of the Figure 5a.

**Figure 6 sensors-19-05033-f006:**
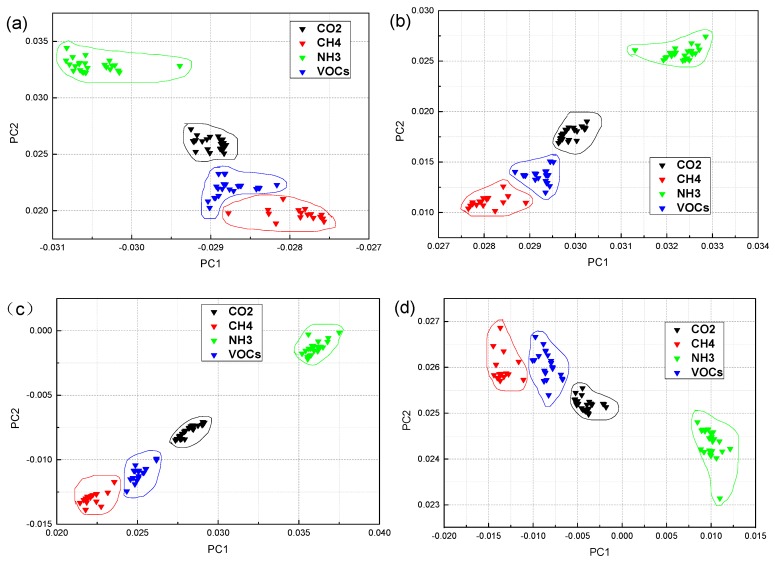
The kernel discriminant analysis (KDA) classification results of the four industrial gas samples under standard deviation: (**a**) c = 10, d = 10; (**b**) c = 10, d = 7; (**c**) c = 10, d = 5; (**d**) c = 10, d = 3; (**e**) c = 7, d = 5; (**f**) c = 15, d = 5.

**Figure 7 sensors-19-05033-f007:**
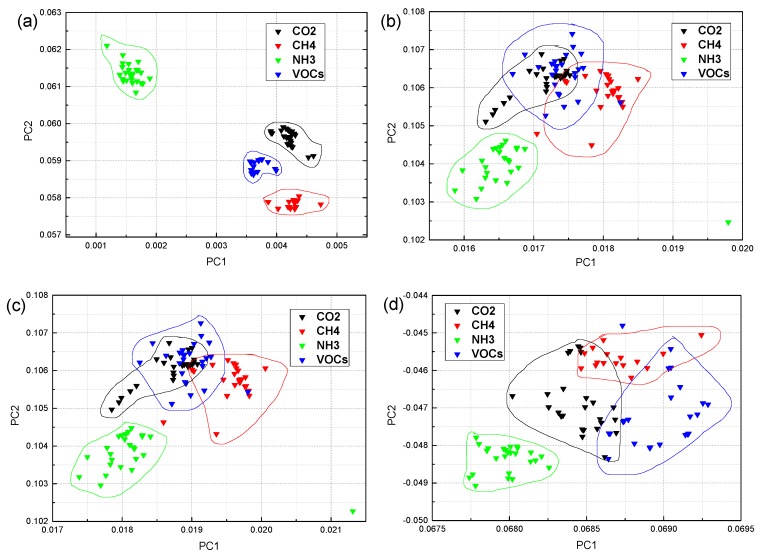
The KDA classification results of the four industrial gas samples with c = 10 and d = 5: (**a**) differential value; (**b**) integral value; (**c**) mean value; (**d**) variance value.

**Table 1 sensors-19-05033-t001:** Electronic nose sensors array and their properties.

Sensor Numbers	Sensor Name	Sensitive Substances	Limit of Detection
Sensor1	W1C	Aromatic	5 ppm
Sensor2	W5S	Ammonia and aromatic molecules	1 ppm
Sensor3	W3C	Nitrogen oxides	5 ppm
Sensor4	W6S	Hydrogen	5 ppm
Sensor5	W5C	Methane, propane, and aliphatic	1 ppm
Sensor6	W1S	Methane	5 ppm
Sensor7	W1W	Sulfur-containing organic matter	0.1 ppm
Sensor8	W2S	Alcohols, carbon chains	5 ppm
Sensor9	W2W	Aromatic, sulfur-containing and chlorine-containing organics	1 ppm
Sensor10	W3S	Methane and aliphatic	5 ppm

**Table 2 sensors-19-05033-t002:** Gas type of samples and parameter setting.

Gas Type	CO_2_	CH_4_	NH_3_	VOCs
Concentration (ppm)	30	30	30	30
Measuring sample	31	31	31	31
Temperature	25 ± 1 °C			
Relative humidity	60% ± 10%			
Sampling time	120 s			
Intake flow	7.8 mL/s			

**Table 3 sensors-19-05033-t003:** Comparison of recognition accuracy for the test gas sample.

Samples	Correctly Identified/Recognition Rate						
	PCA	LDA	PCA + LDA	Support Vector Machine with RBF Kernel (SVM-rbf)	Support Vector Machine with Sigmoid Kernel (SVM-sig)	Back-Propagation (BP)	KDA
CO_2_	100%	83.33%	83.33%	100%	33.33%	66.67%	**100%**
CH_4_	100%	83.33%	100%	33.33%	50%	66.67%	**100%**
NH_3_	100%	100%	100%	50%	33.33%	66.67%	**100%**
VOCs	83.33%	66.67%	83.33%	33.33%	100%	66.67%	**100%**
Average	95.83%	83.33%	91.67%	54.17%	54.17%	66.67%	**100%**

**Table 4 sensors-19-05033-t004:** Comparison of time consumption (in seconds) of the selected algorithms.

Feature Selection	PCA	LDA	PCA + LDA	KDA (*c* = 15, *d* = 5)	KDA (*c* = 10, *d* = 5)
Steady-state	0.156629	0.005640	0.018706	--	--
Standard deviation	0.027845	0.007265	0.026885	0.013880	0.012978
Mean value	0.022214	0.001821	0.023813	0.013002	0.012568
Variance	0.022682	0.001766	0.022306	0.012427	0.012371
Differential	0.019895	0.001792	0.027026	0.012478	0.010387
Integral	0.022618	0.001821	0.029861	0.011811	0.011535

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
