# Peer review of "Classification and Identification of Industrial Gases Based on Electronic Nose Technology"

_sensors, 2019, doi:10.3390/s19225033_

Round 1

Reviewer 1 Report

Abstract: 

KDA abbreviation has not been defined in the text. Moreover please check: "Discriminant" and not "Discriminate".

Please specify results of classification rate, and the c=10, d=5 constant meanings, in the abstract

It is not specified the reduction technique and the classifier used as also the stop criterion utilized

line 44-45: please check syntax: "The data obtained from the detection modules [are] relatively complicated." the sentence lacks of verb. English style and language needs to be checked in all the text.

Materials and methods

line 74-80: which type of sampling has been used ? static or dynamic headspace, is there a sampling vessel ? Is is not clear the "VOCs" which are ? what is the composition of this mixture ?

The authors should specify the program used to obtain results.

line 175-182: this part reports mere speculations please erase or rewrite totally

line 183-191: this part should be reported in the materials and method section more properly

196-197 steady‐state response and mean value these extraction features should be reported in the materials and method section also but with exhaustive descriptions: which sampled signals have been used ? at what time of sampling ?

time consumption is in seconds ?

Conclusions: please report some data about successful classification rate (CR) and time consumed, the description reported results too generic

References: please check literature as important references have been lost of papers using e-nose in numerous applications of odor detection of industrial gases of industrial processed foods:

DOI: 10.1016/j.snb.2018.02.144

DOI: 10.1016/j.foodres.2018.07.067  

DOI: 10.1016/j.jclepro.2016.05.148

DOI: 10.1007/s10151-016-1457-z

Author Response

First, we would like to thank the reviewers and the editor for the positive and constructive comments and suggestions.Please see the attachment.

Reviewer 2 Report

In the paper, the authors presented a novel method for the detection and identification of industrial gases by using E-nose based on different pattern recognition methods. KDA algorithm showed the best classification result. The authors did a interesting research work, and i think the paper can be received after minor revision. The details were as follows:

1 The English of the manuscript should be improved by a native English speakers and well edited (such as the abbreviation of Carbon dioxide is CO2, not CO2 (Line 178)).

2 Why the experimental temperature was 25±1℃, is it the temperature of industrial gases.

3  Carbon dioxide is not a component of organic volatile.

4 The number of training and testing samples should be more clear. The sentence, “A total of four gas samples were measured, and every sample was tested 31 times in parallel. 25 samples were randomly selected for training processing, so there are 100 training samples (4×25 = 100).”, is hard to read.

5 why the sample points was located regularly like a stick in each classification plot.  

Author Response

(The authors gave the same response as above.)

Round 2

Reviewer 1 Report

All the comments I made have been accepted by the authors and replies have been satisfactory, and in its present form the paper is worth of publication